# A COVID-19 Exposure at a Dental Clinic Where Healthcare Workers Routinely Use Particulate Filtering Respirators

**DOI:** 10.3390/ijerph18126481

**Published:** 2021-06-16

**Authors:** Dosup Kim, Jae-Hoon Ko, Kyong Ran Peck, Jin Yang Baek, Hee-Won Moon, Hyun Kyun Ki, Ji Hyun Yoon, Hyo Jin Kim, Jeong Hwa Choi, Ga Eun Park

**Affiliations:** 1Jaw Surgery Center, Department of Oral and Maxillofacial Surgery, Dentistry, Konkuk University of Medical Center, Seoul 05030, Korea; 20180042@kuh.ac.kr; 2Samsung Medical Center, Division of Infectious Diseases, Department of Medicine, Sungkyunkwan University School of Medicine, Seoul 06351, Korea; jaehoon.ko@samsung.com (J.-H.K.); krpeck@skku.edu (K.R.P.); 3Asia Pacific Foundation for Infectious Diseases (APFID), Seoul 06362, Korea; jy34.baek@gmail.com; 4Department of Laboratory Medicine, Konkuk University of Medical Center, Seoul 05030, Korea; hannasis@kuh.ac.kr; 5Division of Infectious Diseases, Department of Medicine, Konkuk University of Medical Center, Konkuk University School of Medicine, 120-1 Neungdong-ro, Gwangjin-gu, Seoul 05030, Korea; kihkdr@kuh.ac.kr (H.K.K.); 20190195@kuh.ac.kr (J.H.Y.); 6Infection Control Office, Konkuk University of Medical Center, Seoul 05030, Korea; 20070345@kuh.ac.kr (H.J.K.); jflower@kuh.ac.kr (J.H.C.)

**Keywords:** COVID-19, SARS-CoV-2, infection control dental clinic

## Abstract

Asymptomatic/mildly symptomatic coronavirus disease 2019 (COVID-19) patients produce a considerable amount of virus and transmit severe acute respiratory syndrome virus 2 (SARS-CoV-2) through close contact. Preventing in-hospital transmission of SARS-CoV-2 is challenging, since symptom-based screening protocols may miss asymptomatic/mildly symptomatic patients. In particular, dental healthcare workers (HCWs) are at high risk of exposure, as face-to-face contact and exposure to oral secretions is unavoidable. We report exposure of HCWs during dental procedures on a mild symptomatic COVID-19 patient. A 32-year-old male visited a dental clinic at a tertiary care hospital. He experienced mild cough, which started three days before the dental visit, but did not report his symptom during the entrance screening. He underwent several dental procedures and imaging for orthognathic surgery without wearing a mask. Seven HCWs were closely exposed to the patient during dental procedures that could have generated droplets and aerosols. One HCW had close contact with the patient during radiologic exams, and seven HCWs had casual contact. All HCWs wore particulate filtering respirators with 94% filter capacity and gloves, but none wore eye protection or gowns. The next day, the patient experienced dysgeusia and was diagnosed with COVID-19 with high viral load. All HCWs who had close contact with the patient were quarantined for 14 days, and polymerase chain reaction and antibody tests for SARS-CoV-2 were negative. This exposure event suggests the protective effect of particulate filtering respirators in dental clinics. The recommendations of different levels of personal protective equipment (PPE) for dental HCWs according to the procedure types should be established according to the planned procedure, the risk of COVID-19 infection of the patient, and the outbreak situation of the community.

## 1. Introduction

Since the first report of severe acute respiratory syndrome coronavirus 2 (SARS-CoV-2) infection in Wuhan, China, in December 2019, there has been a global outbreak of coronavirus disease 2019 (COVID-19) [1]. The typical transmission pathways of SARS-CoV-2 include direct inhalation of droplets and contact between contaminated hands and the nasal, oral, and ocular mucosa [2,3,4,5]. When aerosols are generated in a closed area, aerosol transmission may be another route of infection [6]. Recent studies showed that SARS-CoV-2 can be transmitted by asymptomatic patients who generate few droplets [7,8,9]. Since symptom-based screening protocols may be ineffective at identifying asymptomatic or mildly symptomatic patients, these patients act as hidden transmission foci despite vigorous infection containment measures [7]. Though we implemented reverse-transcriptase polymerase chain reaction (RT-PCR) for SARS-CoV-2 tests on all inpatients at the time of admission, we implemented rather limited infection control policies for outpatients that checked epidemiological association with COVID 19-related symptoms. Moreover, no infection control policy related to the dental field was announced by Korea of even the US Centers for Disease Control and Prevention (CDC).

Healthcare workers (HCWs) at dental clinics are at high risk of unexpected exposure. Regular dental treatments lead to close face-to-face contact with patients not wearing masks. Frequent utilization of vibrating devices may produce aerosols, and body fluids such as blood and saliva can spatter into the eyes. Despite the precautions taken, it is impossible to entirely prevent the production of droplets and aerosols during dental procedures [10]. However, there are several unresolved questions of infection control in dental setting. The most representative question is whether the characteristics of a dental aerosol using high volume evacuation and generated by water or air spray can be seen as the same risk as the aerosol generated during upper airway procedures [11]. Evidence regarding the effectiveness of personal protective equipment (PPE) for dental HCWs during routine patient care during the COVID-19 pandemic is still insufficient. Herein, we report an exposure situation of HCWs during dental procedures on a mildly symptomatic COVID-19 patient with high viral load.

## 2. Methods and Materials

Patient A was a 32-year-old man. On 11 May 2020, he visited a dental clinic at a tertiary care medical center (Konkuk University Medical Center) for a consult prior to orthognathic surgery. After being made aware of SARS-CoV-2 exposure at the dental clinic due to Patient A being diagnosed with COVID 19, we immediately performed thorough contact tracing of the COVID-19 patient. All exposed HCWs, patients, and visitors were identified. We evaluated the degree of contact, such as exposure situation and time, adequacy of PPE, and the presence of any COVID-19-related symptoms. The exposed persons were divided into quarantine (close and unprotected exposure) and active monitoring (distant or protected exposure) groups [12]. Quarantined HCWs were subjected to RT-PCR for SARS-CoV-2 at the start and the end of a two-week quarantine. Serologic tests for SARS-CoV-2 were performed one to two months after the end of quarantine. A fluorescent immunoassay (FIA) kit for anti-SARS-CoV-2 IgM and IgG antibodies and an enzyme-linked immunosorbent assay (ELISA) kit for anti-SARS-CoV-2 total antibody were used. HCWs in the active monitoring group took RT-PCR tests on the day of exposure confirmation and were monitored for the development of COVID-19-related symptoms for two weeks. Informed consent was obtained from the HCWs for donated blood samples. This study was approved by the Institutional Review Board (IRB) of Samsung Medical Center (IRB No. SMC 2020-04-066). We also received consents to obtain blood from all participants. The data was anonymized before the study was conducted.

RT-PCR assay (Seegene Inc., Seoul, Korea) was performed according to the manufacturer’s instructions. The primer and probe sequences were described in a previous study [13]. Each sample was defined as positive if the cycle threshold (Ct) was less than or equal to 35 cycles.

To detect anti-SARS-CoV-2 IgM and IgG antibodies among exposed HCWs, we used a FIA kit (AFIAS COVID-19 Ab assay, Boditech Med Inc., Chuncheon, Korea). The FIA IgM and IgG kit used the automated fluorescent lateral flow immunoassay method, using the AFIAS-6 analyzer system [14]. This assay uses a sandwich immunoassay with a detector SARS-CoV-2 protein (recombinant nucleocapsid proteins with europium chelate). Specimens with a relative cut-off index (COI) value ≥ 1.1 were considered positive. All procedures were performed according to the manufacturer’s instructions.

The ELISA total antibody kit (PCL COVID-19 Total Ab EIA test, PCL Inc, Seoul, Korea) detects total antibody against nucleocapsid protein and receptor-binding domain (RBD) of the spike protein of SARS-CoV-2 using the sandwich immunoassay method. An optical density (OD) ratio < 1.0 was interpreted as negative, ≥0.9 to <1.0 as borderline, and ≥1.0 as positive. All tests were performed according to the manufacturer’s instructions.

## 3. Results

### 3.1. Description of the Exposure Situation at Our Hospital

Although Patient A experienced mild cough three days before the hospital visit, he did not report this symptom during the entrance screening at the hospital. He had a consultation with a dental surgeon and necessary dental procedures as preparation for orthognathic surgery in the dental clinic from 13:30 to 14:45. He underwent chest X-ray, blood sampling, and facial computed tomography (CT) scanning from 14:45 to 15:20. After that, he returned to the dental department for orthodontic treatment. The first step was pre-consultation examination in the oral and maxillofacial surgery department, which included dental impressions, intraoral photographs, extraoral photographs, dental X-rays, wax bite taking, and face-bow transfer. The second step was face-to-face consultation with a professor and one assistant. The third step was orthodontic treatment in the orthodontic department, including a dental bonding procedure and orthodontic wire change. During these dental procedures, Patient A did not wear a face mask. After all the procedures were over, he was given a verbal summary of future treatment procedures. He left the clinic at 16:40. The next morning, the patient developed dysgeusia and his dry cough worsened. He underwent RT-PCR for SARS-CoV-2. On May 13, 2020, he was confirmed to have SARS-CoV-2 infection. The low Ct value on RT-PCR (22.38 for RdRp and 22.52 for E genes) suggested high viral load.

### 3.2. Demographic and Epidemiologic Characteristics of HCWs under Quarantine

A total of 48 persons were identified as exposed, including 15 HCWs. Eight HCWs were quarantined because they had contact with Patient A when he was not wearing a mask. The demographics and epidemiological characteristics of the exposed HCWs are summarized in Table 1. The median age of HCWs under quarantine was 42.5 years old (IQR 30.5–55.75), and four (50.0%) were male. The HCWs consisted of two dentists (25.0%), one nurse (12.5%), four oral hygienists (50.0%), and one radiologic engineer (12.5%). Five of these HCWs worked in the oral and maxillofacial surgery department, two HCWs in the orthodontic department, and one HCW in the radiologic department. 

All procedures are listed in Table 2, which shows how long each task typically takes (time), whether the HCWs talked face-to-face with Patient A (talking), if a droplet was produced during the task (droplet), and whether direct face-to-ungloved hand contact (skin contact) or exposure to oral cavity (oral exposure) occurred.

All HCWs under quarantine wore a particulate filtering respirator with 94% filter capacity (KF94 respirator, approved by the Ministry of Food and Drug Safety (MFDS) of Korea, according to the criteria equivalent to the European Standard for FFP2 respirator), but did not wear eye protection or gowns. Patient A removed his face mask for various dental procedures. Since Patient A was not suspected to have COVID-19, each procedure was performed under normal safety precautions. Hand hygiene before and after patient contact or invasive procedures is an infection control policy that has been of the utmost importance since before the COVID-19 pandemic. This is the infection control policy emphasized at our hospital as well. However, the exposed dental HCWs could not accurately remember whether or not hand hygiene was done. They wore white gowns on scrubs, and these scrubs and gowns were only worn inside the hospital. Ventilation in the treatment room was performed 7.5 times per hour, and 70% of air was recirculated through a high efficiency particulate air (HEPA) filter.

Patient A was not wearing a mask during CT scanning, so the radiologic technician was also quarantined. One HCW complained of myalgia shortly after exposure, but improved within two days.

### 3.3. Laboratory Test Results

All HCWs under quarantine and active monitoring underwent RT-PCR for SARS-CoV-2 four to five days from the date of exposure, and quarantined HCWs were tested again before the end of quarantine. The results of the RT-PCR were all negative. Tests for anti-SARS-CoV-2 antibodies were performed on quarantined HCWs on the 52nd day from exposure. All serologic test results, including FIA IgM and IgG and ELISA total antibody, were negative.

## 4. Discussion

There are currently no data available to assess the risk of SARS-CoV-2 transmission and the prevalence of subclinical seroconversion of SARS-CoV-2 in a dental setting [15]. This is the first descriptive study on real-world exposure to SARS-CoV-2 during dental procedures. In this study, we also investigated seroconversion of SARS-CoV-2 infection in dental HCWs exposed to a mildly symptomatic SARS-CoV-2 patient while using particulate filtering respirators.

Routine application of particulate filtering respirators, such as N95 (US standard), FFP2 (European standard), KN95 (Chinese respirator equivalent to N95), or KF94 (Korean respirator equivalent to FFP2), to protect HCWs from respiratory viruses is controversial. A recent randomized controlled trial resulted in no significant difference between the effectiveness of N95 respirators and surgical masks in preventing influenza infection among participants routinely exposed to respiratory illnesses in a hospital setting [16]. However, faced with the COVID-19 pandemic, use of particulate filtering respirators in high-risk settings should be considered, as there has been some evidence of airborne transmission [17]. Dental practice requires the use of non-disposable dental and surgical instruments, such as handpieces, ultrasonic scalers, and air-water syringes. These instruments can scatter visible droplets that contain water, saliva, blood, microorganisms, and other debris. Surgical masks protect the mucous membranes of the mouth and nose from droplet splash, but they are unlikely to completely protect against the inhalation of virus-containing aerosols [10,15,18,19]. SARS-CoV-2 has been observed to remain viable on inert surfaces for three to twenty-four hours depending on the type of surfaces [20]. Dental unit surfaces and material contamination from patients’ oral fluid could act as sources of infection both for dental HCWs and patients [19]. Fortunately, there has been no reported large-scale outbreak of COVID 19 in dental settings. The US CDC released the following guidelines: dental HCWs should wear a surgical mask, eye protection (goggles, protective eyewear with solid side shields, or a full-face shield), a gown, and gloves during patient care encounters, including those where splashes are not anticipated, in areas with moderate to substantial community transmission. During aerosol-producing procedures, dental HCWs should use N95-equivalent particulate filtering respirators or powered air purifying respirators, where available [15].

At the time of the exposure event reported herein, community transmission of SARS-CoV-2 was not a serious concern in the Republic of Korea. Although a cluster associated with clubs in Itaewon occurred two weeks before, most cases were traced and the number of new daily reported cases was <50 [21]. As a result, guidance for PPE in dental clinics was not as stringent as the CDC has suggested. Although HCWs performed dental procedures or facial CT scanning while wearing only KF94 respirators and gloves, secondary infections did not occur. No HCWs showed seroconversion to SARS-CoV-2. Despite the lack of additional PPE and the high viral load of Patient A, KF94 respirators, gloves, and hand hygiene seem to have prevented viral spread. The KF94 respirator is a particulate filtering respirator with 94% filter capacity, approved by the MFDS of Korea [22,23,24,25]. The MFDS applied equivalent testing criteria as the European Standard for FFP2 respirator (EN 149:2001) to the KF94 respirator (filtration efficiency, using sodium chloride ≥ 94% and paraffin oil ≥ 94%; breathing resistance ≤ 70 Pa (30 L/min); and total inward leakage rate ≤ 11%) [23]. The Korean CDC has recommended that HCWs treating COVID-19 patients wear KF94 respirators to protect themselves against the SARS-CoV-2. However, as most KF94 respirators were designed and manufactured for casual use for protection against fine dust and air pollution before the COVID-19 outbreak, most products use ear loops rather than the headband used in N95 respirators [26]. KF94 respirators with ear loops are more comfortable for daily usage, but they fit looser than respirators with headbands. Currently, most HCWs caring for laboratory-confirmed COVID-19 patients in Korea prefer to use N95 respirators, while KF94 respirators are widely used in routine clinical care and daily life. Although the relative safety of ear loops versus headbands remains controversial, KF94 respirators have advantages in daily use for the protection of HCWs in situations of community transmission of SARS-CoV-2.

Another reason why SARS-CoV-2 spread did not occur at our dental clinic is probably the difference in the degree of droplet and aerosol production depending on dental departments. Moreover, some questions about the aerosol generated during dental procedures remain unresolved. The US CDC regarded the aerosol generated during medical and dental procedures to be an equivalent degree of risk. However, there is an opinion that aerosols generated during dental procedures mainly by water and air-spray cannot be considered to have the equivalent risk as aerosols generated in tracheal or nasopharyngeal procedures because they would substantially dilute the viral presence [11]. 

Furthermore, unlike the periodontal department and the conservative dentistry and prosthesis department, the use of high-speed handpieces is relatively low in the oral and maxilla-facial surgery and orthodontic department, especially for treatments related to orthognathic surgery. Patient A did not undergo typical dental procedures using a high-speed handpiece. All procedures were in preparation for orthognathic surgery. As shown in Table 2, the dental care he received encompassed talking, clinical photos, facing HCWs with mouth open, and dental bonding procedures with a low-speed handpiece only. Thus, there may have been less droplet and aerosol production than during periodontal procedures. 

In the case of non-emergent issues, if a patient suspected to have COVID-19 visits a dental clinic, it is recommended that the dentist defers treatment for at least 14 days. Orthognathic surgery for treating jaw deformities is not a life-threatening condition that must be treated urgently. One study recommended deferral of orthognathic surgery until the COVID-19 pandemic situation has settled [27]. However, these conditions can severely impact the quality of life of the person with the jaw deformity, and surgical interventions cannot be delayed until the end of the pandemic. For dental practice that requires invasive procedures, it is advisable to wear the PPE recommended by the CDC regardless of the patient’s condition. However, as the pandemic continues long term, PPE supplies are becoming increasingly limited. It is difficult and costly to use the recommended PPE for all dental patients [28]. There is an opinion among some that the standard precautions used (surgical masks, gloves, and protective eyewear) for dental procedures before the COVID 19 pandemic are sufficient [11]. However, there was a review article that noted that 38 types of microorganisms could be found floating in the air of dental clinics [29]. Even considering the question of aerosols generated during dental procedures as mentioned above, the possibility of SARS-CoV-2 transmission through dental procedures should always be kept in mind until it is confirmed through accurate research. Nevertheless, in order to preserve resources, different levels of PPE for dental HCWs according to the procedure types should be established according to the planned procedure, the risk of COVID-19 infection of the patient, and the outbreak situation of the community [27]. 

Our study had several limitations. As this was a dental procedure in the department of oral and maxillofacial surgery, production of droplets and aerosols would have been minimal. Only eight HCWs were quarantined. As there was no security camera in the dental clinic, the level of exposure was based on individual recollections. Since this study based its data on a single case report that includes one patient who mainly received a consultation-type appointment without any use of high-speed handpiece, as mentioned above, it might not be representative of all scenarios.

## 5. Conclusions

Among HCWs exposed during dental procedures on a mildly symptomatic COVID-19 patient, there were no SARS-CoV-2 infections. Within the limitation of a case report, the usage of particulate filtering respirators in dental clinic could play a major role in preventing transmission. It is a case showing an effective preventive effect of a high-efficiency mask with ear loop that is relatively easy to wear for a long time (relative to the N95) that could considered for application to the routine practice of a dental clinic. 

## Figures and Tables

**Table 1 ijerph-18-06481-t001:** Demographic and epidemiologic characteristics of HCWs under quarantine and active monitoring.

	Exposed HCWs (n = 15)	HCWs in the Quarantine Group (n = 8)	HCWs in the Active Monitoring Group (n = 7)
**Demographics**			
Age, years	36 (30–52)	43 (31–56)	36 (25–41)
Male sex	6 (40.0)	4 (50.0)	2 (28.6)
Underlying diseases	2 (13.3)	2 (25.0)	0 (0.0)
Occupations			
Doctor	4 (26.7)	2 (25.0)	2 (28.6)
Nurse	1 (6.7)	1 (12.5)	0 (0.0)
Oral hygienist	4 (26.7)	4 (50.0)	0 (0.0)
Radiologic technician	1 (4.7)	1 (12.5)	0 (0.0)
Laboratory technician	5 (33.3)	0 (0.0)	5 (33.3)
**Exposure situation**			
Dental clinic			
Close contact for examination	7 (6.7)	7 (12.5)	0 (0.0)
Distant contact	2 (13.3)	0 (0.0)	2 (28.6)
Outside the dental clinic			
Close contact (blood sample)	1 (6.7)	0 (0.0)	1 (14.3)
Close contact (CT scan)	1 (6.7)	1 (12.5)	0 (0.0)
Distant contact	4 (26.7)	0 (0.0)	4 (57.1)
Patient A’s mask wearing	7 (46.7)	0 (0.0)	7 (100)
**PPE of HCWs**			
Mask/respirator	15 (100)	8 (100)	7 (100)
KF94 respirator	9 (60.0)	7 (87.5)	2 (28.6)
Surgical mask	6 (40.0)	1 (12.5)	5 (71.4)
Gloves	14 (93.3)	7 (87.5)	7 (100)
Gown	0 (0.0)	0 (0.0)	0 (0.0)
Eye protection	0 (0.0)	0 (0.0)	0 (0.0)
**COVID-19-related symptoms ***	1 (6.7)	1 (12.5)	0 (0.0)

Data are expressed as the number (%) of patients or median (interquartile range). * Within 14 days after exposure; Abbreviations: HCWs, healthcare workers; CT, computed tomography; PPE, personal protective equipment; COVID-19, coronavirus disease 2019.

**Table 2 ijerph-18-06481-t002:** Detailed description of dental procedures.

			Time	Talking	Droplet	Skin Contact	Oral Exposure	HCWs (Number)
**Oral and maxillofacial department**	Pre-consultation examination	Dental impression	<15 min	O	O	O	O	Oral hygienist (1)
		Intraoral photo	<5 min	O	O	O	O	Oral hygienist (1)
		Extraoral photo	<5 min	O	X	X	X	Oral hygienist (1)
		Dental X-ray	<5 min	O	X	O	X	Oral hygienist (1)
		Wax bite	<5 min	O	O	O	O	Nurse (1)
		Face bow	<5 min	O	O	O	O	Dentist (1)
	Consultation		<60 min	O	O	O	X	Oral hygienist (2) Dentist (1)
	Dental procedure summary		<10 min	O	O	X	X	Oral hygienists (3)
**Orthodontic department**	Orthodontic treatment	Dental bonding	<30 min	O	O	O	O	Oral hygienist (4) Dentist (2)
		Wire change	<15 min	O	O	O	O	Oral hygienist (4) Dentist (2)

Abbreviations: HCWs, healthcare workers.

## Data Availability

The data presented in this study are openly available in MDPI.

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
