# Peer review of "A COVID-19 Exposure at a Dental Clinic Where Healthcare Workers Routinely Use Particulate Filtering Respirators"

_ijerph, 2021, doi:10.3390/ijerph18126481_

Round 1

Reviewer 1 Report

 Thank you for the opportunity to review your manuscript. This manuscript reported the exposure of health workers during dental procedures on a mild symptomatic COVID-19 patient, which later become worse and followed the health workers and suggested protections and precaution. 

        Although the study discussed an interesting subject, there are few issues that need to be improved. The authors failed to present an elaborated background and gap along with good conclusion. Further insights into the text should be provided as follows:

1- The abstract needs to be improved; conclusion in the abstract is not clear; it's not a conclusion, the take-home message is a suggestion. 

2- Authors, please review your keywords. I suggest using Medical Subject Headings (MeSH).

Introduction:

  • The introduction is brief and is not providing enough information regarding the importance of this case report and gap. Also, the study's novelty is not bold enough. Please add the policy of the country and decisions made to lower the risk of infection for dental health workers.
  • The authors stated "Evidence regarding the effectiveness of personal protective equipment (PPE) for dental HCWs during routine patients care during the COVID-19 pandemic is still in sufficient." This statement needs to be elaborated in the introduction and later in the discussion. 

 Discussion

Although the discussion sets the mood well, it needs information around the policy for providing PPE and precautions around COVID-19. Please clarify the limitation and strength of the study in a more straightforward way.

How does the finding of this study contribute to what we know about the "impacts" of PPE to reduce COVID-19 infection in dentists and other health workers?

The conclusion is very short and superficial. It is more like a suggestion.

Author Response

We deeply appreciate for your kind review and important comments about our manuscript. We did our best to make appropriate revisions according to the reviewer’s comments.

Reviewer 2 Report

The authors have presented an important topic, which will increase the awareness and experience of dental health care professionals.

However, the introduction and literature review are too much concise and requires improvements.

These points should be addressed in the section to explain the importance of the topic as dental health workers (HCWs) are at high risk of exposure. 

1. Periodontal connections to the COVID-19

2. Studies on oral and maxillofacial lesions related to COVID-19 positive patients

3. How dental clinic architecture and environment can affect COVID-19 or viral disease spread

4. Studies on knowledge, awareness of dentists about COVID-19, and cite similar studies

In the materials section please mention the time, date, hospital/clinic name and location of the place where the patient has arrived. 

Author Response

(The authors gave the same response as above.)

Reviewer 3 Report

General comments

The present case report described the screening and management of  15 dental health care workers (HCWs) who had contact with a patient diagnosed with SARS-CoV-2 infection on the day after the dental appointment. The article is well written and enjoyable to follow. I thank the Authors for contributing to the existing literature on the impact of the COVID-19 threat on the clinical activities of dental health care providers. I also invite the Authors to avoid overgeneralizations solely based on a case report that includes one patient who mainly received a consultation-type appointment without any use of high-speed handpiece.

Specific comments

Introduction:

  • Page 2 line 59. There are two periods (..) at the end of the introduction. Reduce to one.

Materials and Methods:

  • I suggest removing the sub-paragraph headings at line 61 (Contact tracing, quarantine, and tests for SARS-CoV-2 infection), line 79 (RT-PCR assay), line 84 (FIA kit for anti-SARS-CoV-2 IgM and IgG antibodies), and line 92 (ELISA total antibody kit).

Results:

  • Page 4 line 143 “Since Patient A was not suspected to have COVID-19, each procedure was performed under normal safety precautions.” Please add a brief description of what is considered normal safety precautions in your clinic. The manuscript already mentioned that the HCWs were not wearing eye protections and gowns. Please add more information, for example if the HCWs performed handwashing before and after every patient contact; what precautions were used on instruments such as camera used for pictures or facebow; what were the HCWs wearing when had contact with the patient (scrubs vs. white coat on normal cloths; long vs. short sleeves); if HCWs’ scrubs were provided and daily cleaned in the clinic or if the HCWs had to bring the contaminated scrubs home; if the rooms had any ventilation or air conditioning devices.

Discussion:

  • It should be stressed that the submitted manuscript based its data on a single case report, which might not be representative of all scenarios.

Conclusions:

  • The sentences “Routine usage of particulate filtering respirators in dental clinic could play a major role in preventing transmission. The effectiveness of PPE for daily patient care during the COVID-19 pandemic should be established depending on the outbreak situation of the community.” appear unsupported by the study design. I suggest rephrasing them to express only what can be concluded based on your study design. For example: “Within the limitation of a case report, the usage of a particulate filtering respirators…”
  • The sentence ”The effectiveness of PPE for daily patient care during the COVID-19 pandemic should be established depending on the outbreak situation of the community.” seems out of context and it should be removed or rephrased.

Author Response

(The authors gave the same response as above.)
